# Terahertz rectification in ring-shaped quantum barriers

Taehee Kang[1], R.H. Joon-Yeon Kim [1,4], Geunchang Choi[1,5], Jaiu Lee[1], Hyunwoo Park[2], Hyeongtag Jeon[2], Cheol-Hwan Park [3] & Dai-Sik Kim[1]

Tunneling is the most fundamental quantum mechanical phenomenon with wide-ranging applications. Matter waves such as electrons in solids can tunnel through a one-dimensional potential barrier, e.g. an insulating layer sandwiched between conductors. A general approach to control tunneling currents is to apply voltage across the barrier. Here, we form closed loops of tunneling barriers exposed to external optical control to manipulate ultrafast tunneling electrons. Eddy currents induced by incoming electromagnetic pulses project upon the ring, spatiotemporally changing the local potential. The total tunneling current which is determined by the sum of contributions from all the parts along the perimeter is critically dependent upon the symmetry of the loop and the polarization of the incident fields, enabling full-wave rectification of terahertz pulses. By introducing global geometry and local operation to current-driven circuitry, our work provides a novel platform for ultrafast optoelectronics, macroscopic quantum phenomena, energy harvesting, and multi-functional quantum devices.

[1] Department of Physics and Astronomy and Center for Atom Scale Electromagnetism, Seoul National University, Seoul 08826, Korea. [2] Division of Materials Science and Engineering, Hanyang University, Seoul 133-791, Korea. [3] Department of Physics and Astronomy and Center for Theoretical Physics, Seoul National University, Seoul 08826, Korea. [4] Department of Physics and Astronomy, Ames Laboratory, Iowa State University, Ames, IA 50011, USA. [5] Department of Energy Science, Sungkyunkwan University, Suwon 16419, Republic of Korea. These authors contributed equally: Taehee Kang, R. H. Joon-Yeon Kim. Correspondence and requests for materials should be addressed to D.-S.K. (email: dsk@phya.snu.ac.kr)

Rectification of light, transformation of oscillating electric and magnetic fields to experimentally observable direct currents, is a key process for ultrafast science[1–4] and energy harvesting[5–7]. Owing to their intrinsic nonlinearity and ultrafast response, tunneling junctions have been widely used in rectification of electromagnetic waves[4,6–10]. In general, a simple picture of one-dimensional potential barrier modulated by light has been successful in describing rectification in tunnel junctions. To achieve efficient rectification from the oscillating light waves using the one-dimensional junction, one needs some suitable asymmetry across the potential barrier. Several efforts have been made such as tailoring time traces of incident pulses[11–13], using electrodes with different work-functions/geometries[5,14], or applying an additional DC bias across the barriers. Recent studies demonstrated precise control of rectified tunneling electrons in sharp nanotips or in metallic nanostructures by illuminating light waves[15–19]. Subwavelength gap structures are exploited to confine incident waves for field enhancement across the one-dimensional barrier[8,20–25], prerequisite for an efficient tunneling process which has a highly nonlinear character[26].

Here, we present a new concept of realizing ultrafast control of tunneling currents by adding a new dimension to the traditional picture of one-dimensional tunneling junctions. We use lateral, ring-shaped tunneling barriers, encasing a metallic island which is surrounded by a metallic plane (Fig. 1a). The total rectified current emerges as a consequence of the contour integration of the tunneling current flux along the perimeter of the loop. By combining nanometer-scale tunneling with two-dimensional macroscopic geometry, we achieve ultrafast full-wave rectification of electromagnetic waves in sub-picosecond time scale that is visualized by femtosecond optical pulses.

## Results

**Terahertz (THz) tunneling in a ring-shaped barrier.** Suppose that an electromagnetic wave impinges on a perfect electric conductor film placed on the $x$–$y$ plane; the incident magnetic field $\mathbf{H}_{inc}$ induces an eddy current per length $\mathbf{K}_{PEC} = \hat{\mathbf{z}} \times (2\mathbf{H}_{inc})$ on the film, which reflects back the incident light and blocks field smearing into the perfect conductor. For a realistic metal film, the amount of induced current is similar to the case of a perfect conductor but now the current per area, $\mathbf{J}(z)$, flows inside the film, whose behavior is characterized by the skin depth[27]. We can simplify the expression of $\mathbf{J}(z)$ by introducing an effective surface current $\mathbf{K} = \int_0^h \mathbf{J}(z)\mathrm{d}z$ where $h$ is the metal thickness[28]. Similar to the perfect conductor case, it can be expressed by $\mathbf{K} \sim \hat{\mathbf{z}} \times \mathbf{H}_0$ where $\mathbf{H}_0$ ($\approx 2\mathbf{H}_{inc}$) denotes the magnetic field outside the top metal surface (i.e., illuminating side). In the presence of sub-wavelength gaps perforated in the metal film, the induced surface current $\mathbf{K}$ charges the gap and applies a potential difference[20,22,29]

$$V(l,t) \propto \int_{-\infty}^t \mathbf{K}(l,t') \cdot \mathbf{n}(l)\mathrm{d}t' \qquad (1)$$

across the tunnel junction positioned at an arc length $l$ and $\mathbf{n}$ denotes the unit vector perpendicular to the contour, directed outward from the loop (Fig. 2a). As the gap width decreases down to the nanometre scale, the induced electric field in the gap is further enhanced by the induced charges of the opposite sides of the gap pulling each other, making enough potential gradient to drive non-negligible tunneling current across the point junctions. Temporal response of the resulting current is straightforwardly determined by the time profile of the incoming electromagnetic field. However, if the tunnel junctions are adjoined together forming a closed-loop, the resulting total current flowing through the loop, $I(t)$, would be a sum over all the point junctions, expressed by the following contour integration,

$$I(t) = \iint_A \mathbf{J}_t(l,t) \cdot \mathrm{d}\mathbf{A} = h \oint_C (\hat{\mathbf{z}} \times \mathbf{J}_t(l,t)) \cdot \mathrm{d}\mathbf{l} \qquad (2)$$

where $\mathbf{J}_t(l, t)$ is the tunneling current flux determined by the electric potential $V(l, t)$ and $\mathrm{d}\mathbf{A} = \hat{\mathbf{n}}h\mathrm{d}l = h\mathrm{d}\mathbf{l} \times \hat{\mathbf{z}}$ and $h$ are the

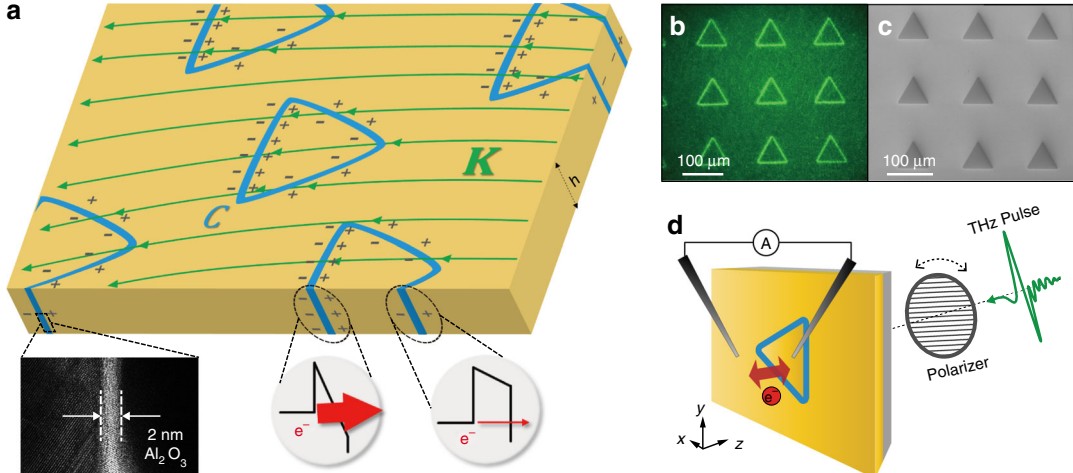

**Fig. 1** Ring-shaped quantum barriers. **a** An external current source generates a surface current (denoted by a vector field **K**) which applies a potential difference across the looped quantum barriers (denoted by contour C) laterally placed on a metallic plane. A TEM picture shows the cross-sectional image of the barrier which is composed of 2-nm-thick Al$_2$O$_3$ layer sandwiched in the gold film. Potentials on the barriers are very sensitive to the angle between the contour and the surface current **K**, reflected in the uneven charge density along each contour element. Electron tunneling probabilities through the barriers are critically affected by the potential difference. **b** An optical microscope image and **c** scanning electron microscope (SEM) image of fabricated triangle loops (side length of 70 μm, gap size of 2 nm). The contrast between the inside and outside of the loops in (**c**) indicates that the ring barriers are well-isolated electrically from the surroundings, affecting the electron emission density during the scanning. **d** Schematic of tunneling current measurement under THz field illumination. The THz field induces eddy currents on the sample surface, which acts as a current source **K** both inside and outside of the triangle as shown in (**a**). Incident polarization of the THz field is controlled by a THz polarizer. Tunneling current through the ring barrier is measured directly by electrical probes

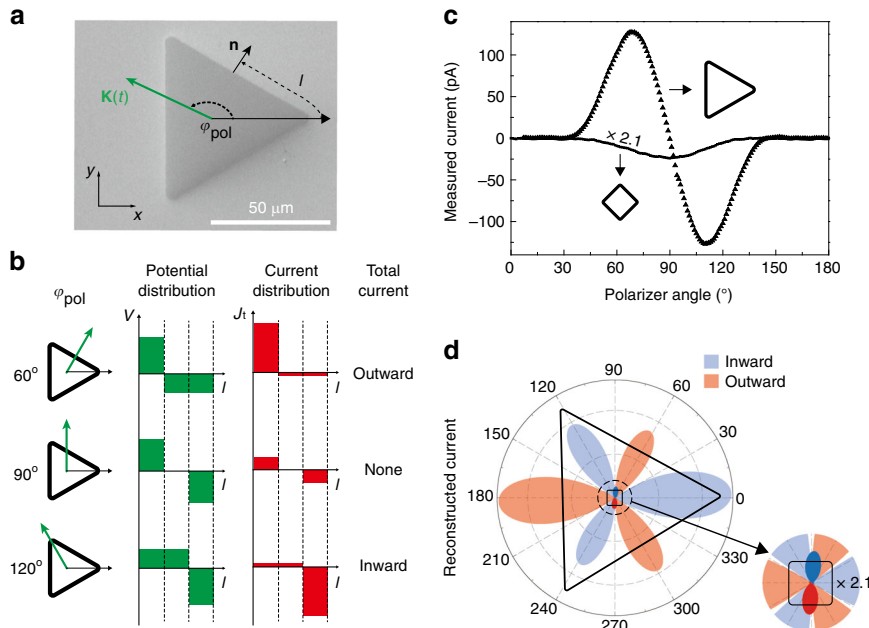

**Fig. 2** Polarization-dependent tunneling in ring-shaped quantum barriers. **a** SEM image of quantum barrier loop in the shape of a triangle. THz-field-driven surface current **K** charges the triangular loop. **b** Schematic description of the polarization-dependent current response in a triangular barrier. The first column shows the incident polarization of the surface current. The second and third columns show the polarization-dependent potential $V(l)$ and tunneling current density $J_t(l)$ (Eq. 3 of Methods) as a function of arc length $l$ (x-axis) defined in (**a**) and polarization (row). The final column shows the total current direction (inward or outward from the loop). The direction of the surface current determines the potential distribution, which in turn drives the tunneling currents across each side of the triangle. The final output current is the contour integral of tunneling currents along the whole quantum barrier loop. **c** Experimentally measured tunneling currents across triangular (side length of 70 μm, perimeter of 210 μm) and square (side length of 25 μm, perimeter of 100 μm) nanogap loops (gap size of 2 nm) are shown as a function of the THz polarizer angle. To directly compare the currents from the two loops of different sizes, a factor of 2.1 (=210 μm/100 μm) is multiplied to the current measured from the square loop. **d** Polar plot of the reconstructed current (red and blue filled curves) obtained from the results in (**c**). The black triangle and square are guides to the eye

area element vector and height of the wall surrounding the contour, respectively. Depending on the incident polarization, the vector relation (integrand of Eq. 2) determines the amplitude and direction of the local current. The contour integration is naturally affected by the loop symmetry. Therefore, the response of the integrated junctions under the rapidly oscillating field is fundamentally different from the response on each junction element, which introduces an entirely new degree of freedom for manipulating the tunneling current.

To realize the concept of the closed-loop barrier and to measure directly the ultrafast currents under electromagnetic pulse excitations, we used two external light sources, a picosecond THz pulse and a femtosecond optical pulse, to trigger currents across the ring-shaped nanogaps fabricated in a 100-nm-thick Au film on a 500 μm thick silicon or quartz substrate[30]. To avoid the unwanted optical absorption by the silicon substrate, we used quartz substrate for optical experiments. The central metallic island is completely isolated from the surrounding metal film by a vertically aligned 2- or 4-nm-thick insulating layer (Fig. 1b, c). By intentionally breaking the inversion symmetry in the new loop geometry, we generate finite total currents via the illumination of THz pulses. The time-integrated total tunneling currents through the barriers are measured directly by attaching electrical probes on the sample surface (Fig. 1d, see Methods for details). A THz polarizer is placed to control the direction of the surface current **K**.

Figure 2b describes the polarization-dependent behaviors of the instantaneous total tunneling current of a triangular loop under a THz field illumination. Electric potential along the contour shows an asymmetric distribution as a consequence of the lack of inversion symmetry inherent in the triangle geometry.

It is interesting to note that the contour integration of the barrier potential affected by the external surface current sources always vanishes independent of the loop shape and loop orientation, automatically eliminating the Ohmic component (see Supplementary Note 1). However, the non-vanishing total current through the barrier naturally emerges for the triangle shape because of the nonlinearity in tunneling current vs. applied potential relation (see Eq. 3 in Methods) together with the triangle's lack of inversion symmetry. Figure 2c shows the measured current responses from triangular and square loops as a function of the THz polarizer angle. The results show strikingly different behaviors depending on the loop geometry. A much higher current flows across the triangular barrier than the square one since the asymmetric potential distribution along the equilateral triangle results in a net tunneling current through the contour while the potential distribution at any point of a square is mostly counterbalanced by its corresponding point across the center, independent of the polarization of the incident pulse. Figure 2d displays polar plots of the tunneling currents for the triangular and square geometries, where the current amplitudes are reconstructed by assuming that the incident field maintains its amplitude for different polarization angles (see Methods for details). The total current vs. polarization angle shows the three-fold rotational symmetry of an equilateral triangle.

**Optical control of THz tunneling currents**. Figure 3a depicts an ultrafast modulation of the THz tunneling current by optical pulses. In this scheme, electric fields of the femtosecond optical pulse and picosecond THz pulse are summed at a specific

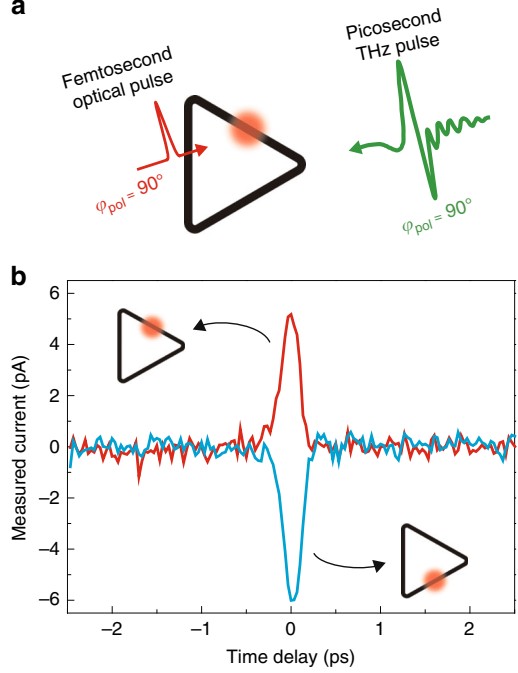

**Fig. 3** Optical manipulation of the THz tunneling current. **a** Ultrafast THz tunneling current manipulation by optical pulse illumination on the ring-shaped quantum barriers. A femtosecond optical pulse generates an additional tunneling current at a specific position of the triangle. **b** Experimental measurement of the time-resolved ultrafast optical gating of the THz tunneling current as a function of the position and time delay of the optical pulse (triangle loop, side length of 70 μm, gap size of 2 nm). The optical pulse modulates the barrier potential spatiotemporally, thereby generating position-sensitive and time-dependent switching signals. Incident polarizations of THz and optical pulses are the same as in (**a**)

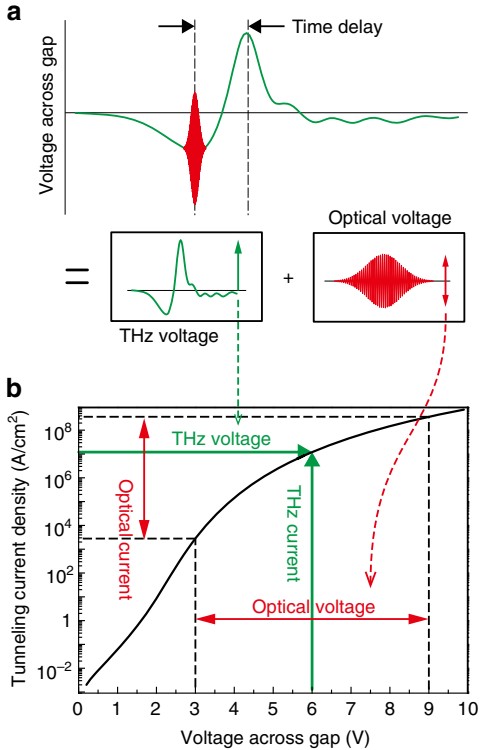

**Fig. 4** Tunneling currents driven by THz and optical fields together. **a** Pictorial description of an electromagnetically applied time-dependent potential across a quantum barrier. A THz field and an added optical field with specific time delay together drive the tunneling current. **b** Tunneling current density calculated by using the Simmons formula (barrier height of 2.2 eV and gap size of 2 nm), describing a non-vanishing optical current under the THz field. Note the log-scale plot of the tunneling current density that shows the extremely nonlinear behavior of the optically induced current

position in the contour. The added optical pulse rapidly distorts the local potential barrier under the quasi-constant THz field, generating an additional, local tunneling current. The resulting current amplitude and direction, driven by the sinusoidal optical field, are critically dependent on the background THz voltage across the gap. The optical current is sensitively affected by the THz field strength at the specific position on the barrier and by the time delay between the THz and optical pulses, thus providing a way to visualize the spatiotemporal dynamics of the THz gap voltage. An interesting aspect of the optically modulated quantum barriers is the position-dependent ultrafast optical gating, as illustrated in Fig. 3b. After the polarization of the THz pulse is intentionally set to generate zero current ($\varphi_{pol} = 90°$, see Fig. 2a for the notation), local modulation of the barrier by an optical pulse breaks the potential balance of the contour and generates ultrafast switching signals of opposite polarities depending on the side of the triangle shone by the optical pulse (see Supplementary Fig. 5 for the $\varphi_{pol} = 60°$ case).

**THz tunneling dynamics revealed by optical pulses**. Under the THz and optical field illumination, the barrier potential is affected by both fields simultaneously. Tunneling current through the barrier is driven by the sum of the light fields, sensitively affected by their temporal field profiles. Figure 4a describes the situation when an optical pulse and a THz pulse are illuminated at the gap together, where they charge the gap and subsequently apply a potential difference across the barrier. The THz pulse profile illustrated in Fig. 4a is the time-dependent potential difference across the gap, which is acquired by using the incident field $\mathbf{H}_{inc}$

(shown in Fig. 1d and Fig. 3a) and Eq. (1). Specifically, the time-dependent voltage is a result of the surface charge accumulated at the gap by the surface current $\mathbf{K}$ induced from the incident field. By integrating the current pulse ($\mathbf{K} \sim \hat{\mathbf{z}} \times \mathbf{H}_0$, where $\mathbf{H}_0$ is the magnetic field just above the metal surface) over time, the applied voltage curve across the gap thus can be described by $V(t) \propto \int_{-\infty}^{t} H_{inc}(t')dt'$ where $H_{inc}$ is the incident magnetic field strength which is proportional to $H_0 \approx 2H_{inc}$ (see Supplementary Fig. 1b, c).

If we compare the measured current under the simultaneous optical and THz excitation with the case of a THz-only excitation, the femtosecond optical field additionally applies a potential difference at a specific time together with the quasi-constant voltage applied by the picosecond THz field (Fig. 4b, see Supplementary Fig. 3 for details). Due to the strong tunneling nonlinearity (Eq. 3 in Methods), the almost sinusoidal oscillation of the optical field nevertheless drives a non-zero current when riding the quasi-static THz field. Figure 5a shows the measured tunneling current under both the THz and optical field illumination by varying the incident optical power. As one can see, the measured current time traces do not strictly follow the THz voltage across the gap (denoted as green dashed line) owing to the tunneling nonlinearity.

Most of the THz tunneling current (without optical illumination) flows only near the intensity maximum of the THz voltage across the gap for a single THz pulse. By analyzing the half widths of the zero-delay peaks of Fig. 5a, we can estimate the timescale of the THz tunneling current pulse. The inset of Fig. 5a shows the

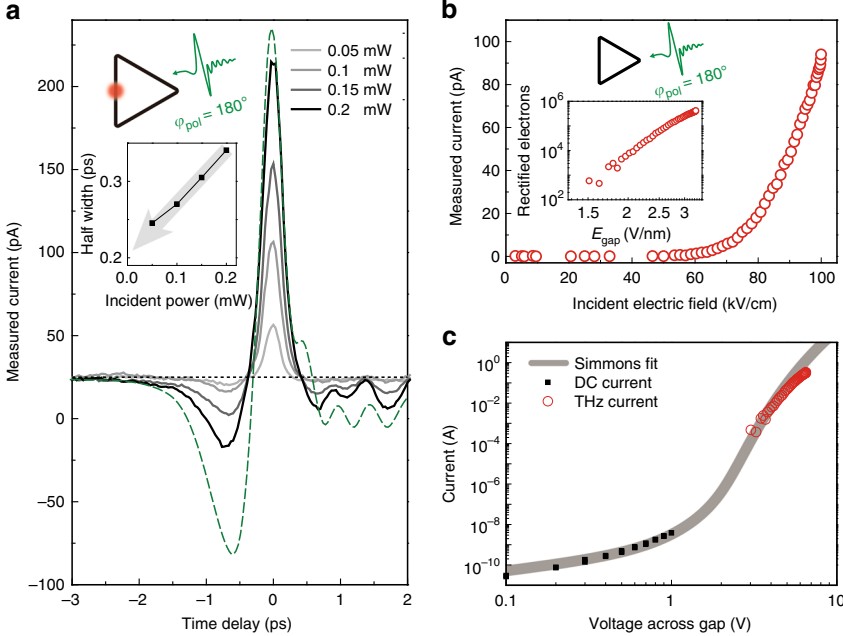

**Fig. 5** THz tunneling dynamics revealed by the optical pulses. **a** Measured tunneling current under the THz and optical pulses (triangle loop, side length of 500 μm, gap size of 2 nm) as a function of the time delay. Legend indicates the incident optical power. Incident polarizations are set to $\varphi_{pol} = 180°$ for both pulses. The green dashed line denotes the voltage across the gap acquired from the incident THz pulse profile (corresponds to the THz pulse shown in (**a**)) and Eq. (1). The black dashed line crossing horizontally at 25 pA is a guide to the eye, indicating the THz current without optical pulse illumination. Inset: extracted half-width of measured currents as a function of incident optical power. Gaussian-fitting was performed for the half width extraction. The grey arrow indicates the experimentally extracted THz tunneling current pulse width. **b** Measured THz current under THz illumination (triangle loop, side length of 70 μm, gap size of 2 nm) as a function of THz field strengths. THz polarization is set to $\varphi_{pol} = 180°$. Inset: total rectified number of electrons per single THz pulse as a function of electric field across the gap. **c** Current–voltage characteristics of the ring barrier. Current data are taken only through a single side of the ring (see Methods for details). DC current and THz current data are tied to the single Simmons curve

extracted half widths as a function of the incident optical power. The half widths become narrower almost linearly as we decrease the optical power; hence, we conclude that the extrapolated half width near zero-optical power (~0.2 ps) is the THz tunneling current pulse width for the given THz voltage pulse profile.

The optical method enables a quantitative analysis of the THz tunneling current vs. voltage relation. By dividing the rectified charge measured from the power-dependent THz tunneling current (Fig. 5b) by the tunneling current pulse width measured from the optical method (inset of Fig. 5a), the THz tunneling current can be quantified with the applied THz voltage across the gap (Fig. 5c, see Methods for details). The resulting THz tunneling current amounts to ~0.3 A, which corresponds to a net flow of ~$4 \times 10^5$ electrons within the 0.2 ps timescale given by a THz gate field that reaches up to ~3.3 V nm$^{-1}$ or a voltage of ~6.5 V across these quantum barriers. Such ampere levels of tunneling currents driven by a THz pulse (~3 V nm$^{-1}$) and additional currents by an optical pulse (~8 V nm$^{-1}$) can be achieved without damaging the barrier during the ultrafast gating with THz and optical pulses[11,12,31]. We confirmed a quantitative agreement between experiment and calculation neglecting thermal effects (see Fig. 5c or Supplementary Fig. 4).

**THz rectification using ring barriers**. A DC bias is another control parameter for manipulating the tunneling current of the ring barriers, enabling ultrafast rectification of electromagnetic waves. Figure 6a shows the total current as a function of the time delay between the two pulses under a DC bias. Here we generated a quasi-monochromatic, multi-cycle THz pulse using spectral filtering, and the polarization is set to $\varphi_{pol} = 180°$ to maximize the response from the left side of the triangle. By sending an optical pulse to the side, we observed the half-wave rectification of an

incoming THz pulse as reflected in the local current. We note that the different noise levels shown in Fig. 6b are due to the unstable current flow at strong DC bias conditions. We observed that the current signals become noisy when the field strength applied by the DC bias under THz illumination reaches ~0.5–1 V nm$^{-1}$ (2–4 V potential difference across the 4 nm gap used in obtaining the results shown in Fig. 6b). This threshold DC field strength also depends on the quality and thickness of the $Al_2O_3$ film. Near this threshold field, the DC current starts to fluctuate and affect the THz current measurement, which makes the noise shown in Fig. 6b.

The integration of these half-wave-rectifying barrier elements along the whole contour results in a full-wave rectification, as illustrated in Fig. 6c. The instantaneous total THz current can be directly visualized by increasing the optical spot size to cover the entire loop. If we set $\varphi_{pol} = 90°$, the potential differences across the upper-right and lower-right sides of the triangle are equal in magnitude and opposite in sign. Now, if an additional DC bias is applied across the loop barrier, this symmetry is broken, resulting in a finite total THz current across the loop barrier independent of the polarity of the THz voltage pulse (Fig. 6d). An optical pulse illuminating both sides provides the instantaneous information on this total THz current (Fig. 6e). Thanks to the unidirectional (i.e., into or out of the loop) current response from all sides of the contour, determined by the DC bias, the THz wave generates a fully-rectified THz tunneling current. We note that if the loop is an ideal equilateral triangle and if the intensity of the optical pulse is constant over the entire contour, the total current across the barrier is zero if the DC bias is zero. But the existence of non-perfect barrier profiles and (or) the slight miss-positioning of the optical pulse (different contribution between upper-right and lower-right sides of the triangle in this case) would result in a non-zero current.

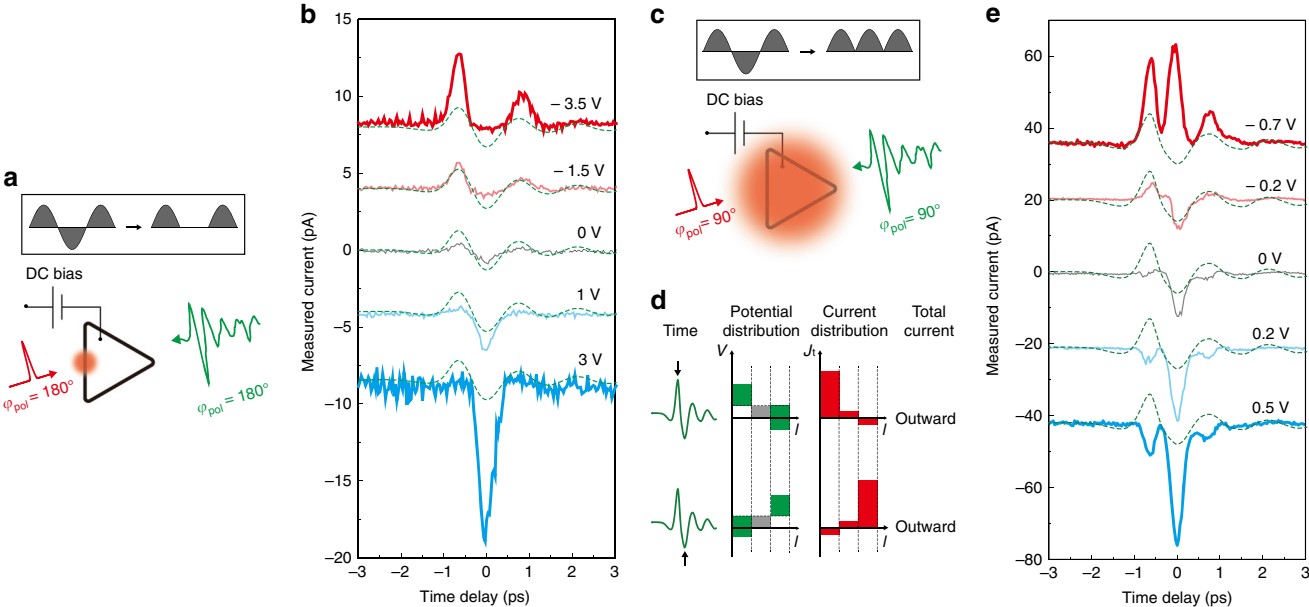

**Fig. 6** Experimental demonstration of ultrafast rectification. **a** Half-wave rectification. Demonstration of a half-rectifying barrier element (side length of 100 μm, gap size of 4 nm). A DC bias imposes potential difference for the contour element, giving a preferred current direction. THz tunneling current is thus rectified on one side of the triangle and an optical pulse selectively probes the temporal dynamics of the rectified current. A multi-cycle THz pulse is used to make a definite oscillating feature of the rectified pulse shape. **b** Demonstration of a half-rectifying barrier element (side length of 100 μm, gap size of 4 nm) for the THz pulses as a function of DC bias. **c** Full-wave rectification. To probe the total THz current, the optical spot is expanded to cover the entire loop. A multi-cycle THz pulse is used. **d** Schematic description of the time-dependent current response in a triangle barrier under a DC bias. The first column shows the specific time of the voltage pulse applied to the barrier, as indicated by black arrows. The second and third columns show the time-dependent potential $V(l,t)$ and tunneling current density $J_t(l,t)$ (Eq. 3 in Methods) as a function of arc length $l$ (x-axis) defined in Fig. 2 (**a**) and time $t$ (row). The final column shows the time-dependent total current direction (inward or outward from the loop). DC bias is described as grey bars and the offset lines in the potential distribution along the circumferential direction. **e** Demonstration of full-rectifying ring barriers (side length of 70 μm, gap size of 2 nm) as a function of DC bias. Dashed green curves in (**b**) and (**e**) show the THz voltage profile acquired from the incident THz pulse profile and Eq. (1). In panel (**b**) and (**e**), the value of the applied DC bias is denoted for each curve and the curves are vertically displaced for clarity

## Discussion

By adjoining the one-dimensional tunneling junctions in two dimensions, the tunneling current flowing across the quantum barriers is determined not only by the external electromagnetic pulse profile (e.g., carrier-envelope-phase), but also by the geometry of the barrier (i.e., lateral symmetry of the ring). Using the proposed method, we can now control the extremely phase sensitive ultrafast tunneling current by modifying the lateral shape of the two-dimensional barrier and by simply changing the polarization of incoming pulses. This concept profoundly widens our modulation technique of ultrafast nonlinear currents, and naturally leads to such unforeseen phenomena as ultrafast full-wave rectifications of THz pulses (see also Supplementary Fig. 6).

Together with the lateral shape of the ring, the optical technique presented in this work can directly reveal the spatiotemporal dynamics of the ultrafast tunneling phenomena. A previous study on THz control of optical photoemission from a nanotip[17] showed that photoelectrons follow the time trace of the THz field applied at the tip, demonstrating a temporal control of photoemission by THz pulses. Our method exploits the small beam size of the optical pulses and allows the spatiotemporal control of THz tunneling currents by optical fields. This new combination method enables visualization of the spatiotemporal dynamics of the THz tunneling current in the ring barrier, position-sensitive optical gating of THz pulses, and quantification of THz tunneling timescale across the barrier. These newly developed techniques will have a deep impact on the research community working on ultrafast phenomena.

The peak current density driven by the THz pulse in our data of ~4.3 MA cm$^{-2}$ at a field strength of ~3.3 V nm$^{-1}$ is similar to the current densities and field strengths across one-dimensional nanogap junctions used in previous studies. However, the amount of measured current in our experiment is much larger since we utilized the whole loop (the total loop area of ~$7 \times 10^6$ nm$^2$ for a triangle barrier whose side length is 70 μm and height is 100 nm) compared with a single point tunnel junction (the total junction area of ~80 nm$^2$ or less, such as STM tips or bowtie-shaped nanogaps) used in most other studies.

In conclusion, we demonstrated a highly nonlinear light-matter interaction, taking advantage of the two-dimensional, lateral geometry of closed-loop quantum barriers, whose lack of macroscopic inversion symmetry plays a vital role in the ultrafast optoelectronics. The ring-shaped quantum barriers introduce a new control method for tunneling currents, further enriched by the femtosecond optical excitation at designated areas and time delays. By implementing the contour-integral concept and lateral symmetry into the conventional one-dimensional tunneling, we realized a multifunctional quantum device, providing a platform for optical transistors, ultra-high bandwidth communications and wireless energy conversion.

## Methods

**Sample preparation**. The ring-shaped nanogaps are prepared by atomic layer lithography technique[30]. On a 500-μm-thick quartz (or low conductivity silicon) substrate, AZ5214 image reversal photoresist was spin-coated at 4000 rpm for 60 s, then prebaked at 90 °C for 60 s. The resist-coated sample was exposed to UV light (350–450 nm wavelength, beam intensity about 20 mW cm$^{-2}$, MIDAS mask

aligner) under photomask for 6 s. Then post-baking was performed at 120 °C for 120 s. After the second exposure to the same UV light for 40 s, samples were immersed in MIF500 developer solution for 30–60 s depending on the pattern size. After the deposition of 100-nm-thick Au layer by an e-beam evaporator and a subsequent lift-off process using acetone with a 1 min sonication, a 2- or 4-nm-thick alumina layer was coated by atomic layer deposition, which determines the nanogap size. After this, another Au layer of 100 nm was deposited directly onto the previous pattern. Finally, an adhesive tape was applied to the sample to planarize the surface. The side lengths of triangular and square patterns were varied from 10 to 500 μm depending on the experiments.

**Generation and detection of THz waves.** In this work, a broadband high power THz source is used, whose spectrum ranges from 0.1 to 3 THz (Supplementary Fig. 1). Single-cycle THz pulse is generated by a prism-cut lithium niobate (LiNbO₃) crystal via pulse-front-tilted optical rectification[32]. Optical pulse (amplified 1 kHz Ti:sapphire laser: wavelength of 800 nm, pulse energy of 5.3 mJ, pulse width of 35 fs, Spitfire, SpectraPhysics) was divided by a 99:1 beam splitter for THz pulse generation (99%) and time-resolved tunneling measurements (1%). Generated THz beam was guided by a series of off-axis parabolic mirrors. The incident field strength was controlled by a pair of wire grid polarizers. For the polarization resolved measurements, we used a single polarizer. Transmitted THz field through the sample is detected in time-domain through electro-optic (EO) sampling[33]; A 200-μm-thick ZnTe crystal and subsequent quarter wave plate with a pair of photodiodes probe the THz field. Supplementary Figure 1b, c show the EO sampling signals of the generated THz waveforms and their time-integrated pulses (voltage across the gap), showing ~1 ps of single-cycle THz pulse (used in Figs. 1–5) and multi-cycle THz pulse (used in Fig. 6). Supplementary Figure 1d, e show the measured THz transmitted amplitude of triangle nanogaps, normalized by the substrate transmitted amplitude.

**Calculation of tunneling current.** We used a full integral expression of the Simmons formula[34,35] to calculate tunneling currents and model the experimental data in this work. Tunneling current density $J_t(V)$ across the one-dimensional barrier can be expressed by

$$J_t(V) = \frac{4\pi m e}{h^3}\left[\int_0^\eta D(E)(\eta - E)dE - \int_0^{\eta - eV} D(E)(\eta - eV - E)dE\right] \quad (3)$$

where $m$ is the electron mass, $e$ the electron charge, $h$ the Planck constant, $\eta$ the Fermi level of the metal, $V$ the applied voltage across the barrier, $E$ the tunneling electron energy and $D$ is the electron tunneling probability factor based on the WKB approximation

$$D(E) = \exp\left(-\frac{4\pi \Delta s\sqrt{2m}}{h}\sqrt{\eta + \bar{\varphi} - E}\right) \quad (4)$$

and the mean value of the effective barrier height

$$\bar{\varphi} = \frac{1}{\Delta s}\int_{s_1}^{s_2}\left(\varphi_0 - \frac{eVx}{s} - \frac{1.15\lambda s^2}{x(s-x)}\right)dx \quad (5)$$

where $\varphi_0$ is the rectangular barrier potential height, $s$ the thickness of the insulating layer, $\lambda = e^2\ln2/16\pi\epsilon s$ ($\epsilon$ = dielectric constant of the barrier), and $\Delta s = s_2 - s_1$ is the effective barrier width where the limits $s_1$ and $s_2$ are given by the real roots of the cubic equation $\varphi(x) = \varphi_0 - eVx/s - 1.15\lambda s/x(s-x) = 0$.

**Tunneling current measurements.** Tunneling currents are measured by attaching electrical probes directly on the sample surface. To safely attach a metallic probe inside the pattern without damaging the sample surface, an electrochemically etched tungsten wire was used as the probe tip, with the tip radius of curvature of ~1 μm. One probe is connected to a Keithley 2450 sourcemeter to apply a DC bias and the other probe is connected to a current preamplifier followed by a lock-in amplifier synchronized to the laser repetition rate (1 kHz). Both tips are positioned on the opposite side to the illuminating direction to avoid blocking the incident THz beam. All electrical apparatuses were on the same ground.

To check the validity of the tunneling process in our ring barrier structure, we performed current-voltage (IV) measurement under for two different cases: application of a DC bias voltage and of THz pulses. Black-dots in Fig. 5c shows a DC IV measurement, demonstrating a general exponential behavior of a one-dimensional tunneling junction. From the curve fitting process based on the Simmons formula, we extracted the barrier potential of the Al₂O₃ layer used in our experiments (~2.2 eV) and layer thickness (~2 nm), the latter of which is also estimated from the TEM picture analysis (Fig. 1a). To compare the measured DC IV data and the THz measurements on an equal footing, we considered the DC current flowing through one side of the triangle. Thus we divided the DC current by three; assuming an equivalently distributed DC current through all three sides.

Next we measured tunneling current as a function of the incident THz field strength, as shown in Fig. 5b. A pair of wire-grid polarizers were used to vary the THz field strength with the incident polarization set to $\varphi_{pol} = 180°$. The resultant THz electric field applied across the gap (x-axis, inset of Fig. 5b) is estimated by the

Kirchhoff integral formalism[20,36]: The measured THz transmitted amplitude ($t \sim 3.5 \times 10^{-3}$, Supplementary Fig. 1d, e) and the coverage ratio of our sample ($\beta \sim 1.07 \times 10^{-5}$) gives the field enhancement factor ($t/\beta$) of ~348. With the measured field enhancement factor, the resulting gap field was estimated by $E_{gap} = E_{inc} \times$ (field enhancement), where $E_{inc}$ is the incident field strength. And we converted the measured current data to the total rectified charges (y-axis, inset of Fig. 5b). The bandwidth of our current preamplifier is 2 kHz (with sensitivity of 10 nA V⁻¹) which is much lower than that of the tunneling current pulse (expected to be higher than 1 THz from the tunneling nonlinearity). In this case, the measured signal in the lock-in amplifier is proportional to $q_{THz}/\tau_{rep}$, where $q_{THz}$ is the total rectified charges under a single THz pulse,

$$q_{THz} = \int_0^{\tau_{rep}} I(t)dt \quad (6)$$

where $\tau_{rep}$ is the pulse to pulse separation time and $I(t)$ the instantaneous total tunneling current in Eq. (2). To find a quantitative relation between the output of the lock-in amplifier and the total rectified charges, we modeled the time trace of the current preamplifier response $I_{amp}(t)$ (inset of Supplementary Fig. 1a) by Fourier expansion

$$I_{amp}(t) = \sum_p C_p \cos\left(2\pi p \frac{t}{\tau_{rep}}\right) \quad (7)$$

where $p$ is an integer and $C_p$ is the corresponding Fourier coefficient. Thus $C_1$ is directly related to the lock-in amplifier output which is synchronized to the pulse repetition rate. Our aim is to find a relation between $q_{THz} = I_0\tau$ (integration of $I_{amp}(t)$ over time, assuming $I_{amp}(t)$ as a square pulse) and $C_1$, where $I_0$ is the maximum of $I_{amp}(t)$ and $\tau$ is the current pulse width of $I_{amp}(t)$. By multiplying cos ($2\pi t/\tau_{rep}$) and integrating in $[-\tau_{rep}/2, \tau_{rep}/2]$ for both sides, $C_1$ can be expressed by

$$C_1 = \frac{2I_0}{\pi}\sin\left(\frac{\pi\tau}{\tau_{rep}}\right) \quad (8)$$

This can be further reduced to $C_1 = 2I_0\tau/\tau_{rep}$ by assuming $\tau \ll \tau_{rep}$ (while this assumption is not rigorously valid under the current preamplifier bandwidth of 2 kHz presented here, we confirmed the same signal level for a larger bandwidth of 20 kHz by decreasing the sensitivity of the preamplifier under a large signal condition, thus justifying our assumption). Considering the root-mean-square output of the lock-in amplifier ($C_1/\sqrt{2}$ = measured current), we can write the total rectified charges for the single THz pulse by

$$q_{THz} = (\text{measured current}) \times \tau_{rep}/\sqrt{2} \quad (9)$$

We finally converted the measured current to the instantaneous tunneling peak current via dividing $q_{THz}$ by the measured tunneling current pulse width and taking the current through one side of the triangle (Fig. 5c) by considering the current distribution (Fig. 2b).

**Modeling polarization-dependent response of tunneling currents in ring-shaped barriers.** Under the THz field illumination, tunneling current through the triangle barrier sensitively depends on the incident polarization. We modeled the polarization-dependent total instantaneous current $I_{total}$ by the following equation (Supplementary Fig. 2a),

$$I_{total} = \sum_i a_i J_t(E_{inc}\sin(\theta_i)) \quad (10)$$

where $a_i$ is the asymmetry factor for each side of our quantum barrier figure, $J_t(E)$ the tunneling current density as a function of the field strength $E$, $E_{inc}$ the incident field strength, $\theta_i$ the angle between the THz induced surface current and the figure side, and $i$ runs through the number of sides of the figure. Here, the asymmetry factor reflects the non-uniformity of a realistic barrier in our sample. The black line of Supplementary Fig. 2b shows the raw data presented in Fig. 2c for a triangle (Malus' law). Due to the slight non-uniformity of each side, $a_i$ values of 1, 0.75 and 0.67 were used for the fitting process. Using these parameters and taking $J_t(E) = J_t(E_{inc})$, we can describe the full 360 degrees polarization responses as shown in Supplementary Fig. 2b. Together with the fittings for the square-shaped-barrier sample, the reconstructed polar plots of Fig. 2d are obtained.

## Data availability
The data that support the findings of this study are available from the authors on reasonable request.

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

## Acknowledgements

We thank Jisoon Ihm for helpful discussion. This work was supported by the National Research Foundation of Korea (NRF) grant funded by the Korea government (MSIT: NRF-2015R1A3A2031768) (MOE: BK21 Plus Program-21A20131111123).

## Author contributions

R.H.J.K. and D.S.K. conceived the study. T.K., R.H.J.K. and G.C. performed experiments and analyzed the data. T.K., R.H.J.K. and J.L. fabricated samples. H.P. and H.J. performed ALD for Al$_2$O$_3$ deposition on the samples. T.K. and D.S.K. wrote the manuscript. R.H.J.K., G.C., C.H.P. and D.S.K. commented on the manuscript.

## Additional information

**Competing interests:** The authors declare no competing interests.

