## [Peer Review File · Nature Communications]

Reviewers' comments:

Reviewer #1 (Remarks to the Author):

The manuscript reports an investigation of tunneling electrons induced THz and DC biased fields in several nano-gaped systems that are fabricated to exhibit closed tunneling barriers and broken symmetries. Authors present evidence for the detection and manipulation of ultrafast currents by position- and polarization-selective optical/THz pumping. Using the concepts from global geometry and local perturbation in a circuitry model, authors are able to qualitatively reproduce the various aspect of their observations and argue the consequence towards "ultrafast optoelectronics, energy harvesting and multi-functional quantum devices."

My general impression for this work is that it contains solid experimental/modeling results and the analysis is carefully done. That is being said, however, it lacks the fundamental insights to quantum tunneling physics and/or technical demonstrations that exceed what have been achieved so far in the community, e.g., in references 12-14 in the manuscript. The nano-gaped structure and tunneling current measurement have been achieved in the literature. It is certainly an interesting direction to increase the knowledge and authors are in the very good position to do these experiments. Yet it is just short of the breakthrough, both conceptually and technically, which could motivate the community to follow. There are a few other comments that could be useful to improve the manuscript.

(1) I notice the THz pulses used are quite asymmetric in the time domain (i.e., non-zero when integrating in time), e.g., in Figs. 2a and 3. Could authors comment on if such effect contribute to the tunneling current?

(2) This paper is very hard to understand at first since the main text is very short yet with very long supplementary. This seems to be written for some other journals with a restricted length. I would highly recommend to move some of the supplementary information to the main text and reconstruct the logic flow to make these nice results understandable. I think both the paper and audience will be greatly benefited with such improvement.

(3) Equation 2 in the main text looks suspicious and please double check the A and I.

Reviewer #2 (Remarks to the Author):

The authors report ultra-fast rectification of the THz radiation on thin ring-shaped triangular tunnel junctions created at the interface between two gold regions. Coupling of the gold film endowed with such junctions to impinging picosecond pulse of the electromagnetic radiation induces Eddie currents in the film that causes potential redistribution at the tunnel interface between the film and the gold island. In particular, the potential difference across the junction can be so strong that it ensures finite tunnelling current between the gold film and the island. Furthermore, thanks to the triangular asymmetric shape of the island the built-up potential depends on the relative orientation between the surface current direction and the vector orthogonal to the triangle sides that defines the predominant direction of the tunnel current and leading to a finite dc signal measured between the film and the island (as oppose to the symmetric square-shaped loop-junction where the total tunnel current will be cancelled by the symmetry). Moreover, the authors demonstrated the control of the tunnelling current by femtosecond optical pulse and dc bias. The approach proposed by authors to control THz fields is novel, well explained and will be of interest for Nature Communication readers. I would recommend the publication of this manuscript in Nature Communications upon addressing the following comments and questions.

1. First of all, the revision of the first figure is needed. In particular, it takes a while to realize where the tunnel current flows. It became clear only after reading the supplementary information and related papers on the fabrication of such junctions. Authors may want to revise Fig. 1b providing the illustration of the junction crosssection and demonstrate the measurements schematic such as it is done in Fig. S3.
2. In all figures, authors illustrate the triangles colour-coded from yellow to green but never address what these colours mean? Do they correspond to the potential build-up upon coupling to the picosecond pulse? If yes, then I would assume it to be different for different polarization angles ϕ_{pol} although it is the same in all the panels of Fig. 1c. Please elaborate.
3. In Fig. 2b authors report the tunnel current induced by gating with femtosecond optical pulse when the polarization of the THz pulse is adjusted to give zero net flow. Authors provide the measurements at two symmetric positions of the optical laser spot. Is it possible to map the full loop (or even the full metallic island) in this way? Do the colours of the schematic triangle represent this distribution? If yes, please report the colour-map scalebar. What happens at other angles of the THz pulse polarization. It can be interesting for the readers to see few maps of the potential (or current) distribution if possible.
4. In all the figures authors report the current in arbitrary units while the measurements are performed by a standard lock-in technique. Would that be possible to report the actual current flowing across the junction as it is done in fig. S3? With that respect, data in Fig. 3 have different noise level for different DC bias voltages. Authors may want to elaborate on this and explain if the data is normalized to some value or shown as-measured. At least, please provide the axis-ticks on the y-axis in all the figures.
5. In the final part of the paper, the authors report the modulation of the tunnelling current upon changing the DC bias voltage across the junction. In Fig. 3 a and b the authors provide the green dashed line which "denotes" the THz voltage profile across the barrier? How is this curve obtained?
6. Please provide the description of insets in Fig. 3 (green wave-like schematic).

Reviewer #3 (Remarks to the Author):

This is a nice article that was quite enjoyable to read. The subject matter is of interest to many as the rectification of high-frequency radiation is becoming a hot topic.

There are a few points that I missed in the description and measurement that I think the authors should include for a more complete paper. First, the thickness of the metal film is not listed, at least I did not notice it. This is an important parameter and should be included. If thin, the metal may not be optically opaque at either the THz or 800nm optical frequency of the two pulses. Second, although a reference is listed (ref. 16) where a process is shown, it is unclear in this case if the same materials are used. Is the substrate again silicon, and is the substrate left on for the measurements? If so, this should also be mentioned in the theory and numerical section as the incident half plane and back half plane (if the metal is thin) are different materials.

The thickness and substrate are also of interest when considering the measurement. I would like to ensure that all other factors are not factors when the rectification measurement is performed. One of these is clearly localized heating when these pulses are incident. The peak tunneling current of 0.2A (supplemental page 4) is large over such a small area as these triangle geometries

and nm-scale tunnel widths, even if brief in duration. Is this temperature change effect considered?

Also, if the metal thickness is not several skin depths thick at the optical pulse wavelength of 800nm then is the effect of its absorption in the silicon considered in the device. Where would this energy go in the device under the test conditions.

These are largely engineering issues rather than physics, however the paper would be stronger if addressed.

RESPONSE TO REFEREE COMMENTS

Reviewers 2 and 3 consider that the research reported in our paper is novel and of broad interest to the general readers of *Nature Communications*. Specifically, Reviewer 2 commented that “The approach proposed by authors to control THz fields is novel, well explained and will be of interest for Nature Communication readers” and recommended publication upon revising our paper following several comments. Reviewer 3 commented that “This is a nice article that was quite enjoyable to read. The subject matter is of interest to many as the rectification of high-frequency radiation is becoming a hot topic” and, “for a more complete paper,” raised questions which “are largely engineering issues rather than physics, however the paper would be stronger if addressed.”

Reviewer 1 commented that our paper “contains solid experimental/modeling results and the analysis is carefully done” but he or she considers that that “it lacks the fundamental insights to quantum tunneling physics and/or technical demonstrations that exceed what have been achieved so far in the community, in references 12-14.” We thank the reviewer for allowing us to realize that we were not good in revealing the conceptual and technical breakthroughs of our study.

Among other useful comments from Reviewer 1, we especially appreciate the comment that “This paper is very hard to understand at first since the main text is very short yet with very long supplementary. This seems to be written for some other journals with a restricted length.” In fact, our manuscript was initially prepared for submission to Science as a one-page Brief Report. But we fully agree with Reviewer 1 that we had better “move some of the supplementary information to the main text and reconstruct the logic flow to make these nice results understandable” and that “both the paper and audience will be greatly benefited with such improvement.”

We have (1) significantly revised our manuscript to reveal the conceptual and technical breakthroughs of our study, (2) moved a significant amount of material from supplementary information to the main manuscript and to Methods, and (3) faithfully followed all other reviewer comments and revised our manuscript accordingly. With these revisions, we hope that our paper can now be accepted for publication in *Nature Communications*.

DETAILED REPLY TO REVIEWER 1

We thank Reviewer 1 for the important comments and suggestions which significantly improved our manuscript. Blue parts are from the referee comments.

My general impression for this work is that it contains solid experimental/modeling results and the analysis is carefully done. That is being said, however, it lacks the fundamental

insights to quantum tunneling physics and/or technical demonstrations that exceed what have been achieved so far in the community, e.g., in references 12-14 in the manuscript. The nano-gaped structure and tunneling current measurement have been achieved in the literature. It is certainly an interesting direction to increase the knowledge and authors are in the very good position to do these experiments. Yet it is just short of the breakthrough, both conceptually and technically, which could motivate the community to follow.

We thank the reviewer for the compliment on our work that it contains solid experimental / modeling results and the analysis is carefully done. Moreover, we fully agree with the reviewer that “the nano-gaped structure and tunneling current measurement have been achieved in the literature.” We recognize that the previous version of our manuscript was written in such a way that the novelty of our work was not clearly revealed in the context of what had been achieved before. Indeed, there have been several studies on the tunneling current measurements using external electromagnetic waves and nanostructures. We however note that all previous studies (e.g., references 12-14 of the original version of our manuscript) employed quantum tunneling across a *one-dimensional* junction (such as bowtie-shaped nanogaps or STM tips). With such one-dimensional junctions, it was inevitable to delicately control the exact time profile of the ultrafast electromagnetic oscillation (such as carrier-envelope-phase control technique) to modulate the currents. On the other hand, the method provided in our work, controlling the quantum tunneling driven by light pulses using *two-dimensional* ring-shaped barriers, has the following key advances compared with previous studies.

(1) **By adjoining the one-dimensional tunneling junctions in two dimensions**, the tunneling current flowing across the quantum barriers is determined not only by the external electromagnetic pulse profile (e.g. carrier-envelope-phase), but also by the geometry of the barrier (i.e. lateral symmetry of the ring). Using the proposed method, we can now control the extremely phase sensitive ultrafast tunneling current by modifying the lateral shape of the two-dimensional barrier and by *simply* changing the polarization of incoming pulses. This *conceptual* breakthrough naturally leads to *technical* breakthroughs, profoundly widening our modulation technique of ultrafast nonlinear currents to such unforeseen phenomena as **ultrafast half- and full-wave rectifications** of terahertz pulses.

(2) Together with the lateral shape of the ring, the optical technique presented in this work can directly **reveal the spatiotemporal dynamics of the ultrafast tunneling phenomena**. There was a previous study on THz control of optical photoemission from a nanotip [Ref. 15 of our manuscript] showed that photoelectrons follow the time trace of the THz field applied at the tip, demonstrating a *temporal* control of photoemission by THz pulses. Our paper presents a significant advancement in this optical-and-THz-waves combination method. Our method exploits the small beam size of the optical pulses and allows the *spatiotemporal* control of THz tunneling currents by optical fields. This new combination method enables (i) **visualization of the spatiotemporal dynamics of the THz tunneling current** in the ring barrier, (ii) **position-sensitive optical gating** of terahertz pulses, and (iii) **quantification of THz tunneling timescale** across the barrier. These newly developed techniques will have a deep impact on the research community working on ultrafast phenomena.

(3) By exploiting the much larger area of the loop tunneling barriers compared with that for point junction, **we can generate a relatively large current by just a single electromagnetic pulse, which could eventually lead to much more efficient and cheaper signal processing schemes.** The peak current density driven by the THz pulse in our data of $\sim 4.3 \text{ MA/cm}^2$ at a field strength of $\sim 3.3 \text{ V/nm}$ is similar to the current densities and field strengths across one-dimensional nanogap junctions used in previous studies. However, the amount of measured current in our experiment is much larger since we utilized the whole loop (the total loop area of $\sim 7 \times 10^6 \text{ nm}^2$ for a triangle barrier whose side length is $70 \text{ }\mu\text{m}$ and height is 100 nm) compared with a single point tunnel junction (the total junction area of $\sim 80 \text{ nm}^2$ or less, such as STM tips or bowtie-shaped nanogaps) used in most other studies.

In summary, we anticipate that the conceptual breakthrough of our study, finding out the crucial role of the macroscopic geometry of tunneling junctions, and several technological advancements based thereupon would eventually lead to the realm of strong light-matter interactions, creating an entirely new set of applications such as ultra-high bandwidth technology, optical transistor, and light energy converter.

We have fully incorporated the above discussion (clarification of the breakthroughs of our study in light of what has been done before) into our manuscript. We thank Reviewer 1 for this significant improvement of our paper.

(1) I notice the THz pulses used are quite asymmetric in the time domain (i.e., non-zero when integrating in time), e.g., in Figs. 2a and 3. Could authors comment on if such effect contribute to the tunneling current?

This is an important, potentially confusing point of our manuscript. It is known that [Kim et al., Phys. Rev. Lett. **84**, 3210 (2000)] the field from a pulse when integrated over time at a fixed position in the far-field regime should vanish. Actually, the integration over time of a THz pulse used in our experiment vanishes. The pulse profile in Fig. 3a (Fig. 2a of the previous manuscript) is the incident **THz field** profile, measured by EO sampling method (details are presented in Methods and Supplementary Fig. 1). This field would generate an eddy current (effective surface current $\mathbf{K} \sim \hat{\mathbf{z}} \times \mathbf{H}_0$, where $\mathbf{H}_0 (\approx 2\mathbf{H}_{\text{inc}})$ is the magnetic field above the top metal surface) on the metal film and makes a potential difference across the gap determined by Eq. 1 of the main manuscript. Thus, the “**voltage across the gap**” shown in Fig. 4a is acquired by time integrating the THz field and converges to zero after the THz pulse passes by. We added this discussion on the THz voltage pulse profile to our revised manuscript.

Panels from Supplementary Fig. 1. Left column shows original THz pulse, and right column shows the transformed THz gap voltage profile by integrating left curves in time.

(2) This paper is very hard to understand at first since the main text is very short yet with very long supplementary. This seems to be written for some other journals with a restricted length. I would highly recommend to move some of the supplementary information to the main text and reconstruct the logic flow to make these nice results understandable. I think both the paper and audience will be greatly benefited with such improvement.

We fully agree with the reviewer. Faithfully following the advice, we moved a significant amount of material (figures and paragraphs) from Supplementary Information to the main manuscript and to Methods.

(3) Equation 2 in the main text looks suspicious and please double check the A and l.

We have double checked the equation and found no errors. In order to prevent any error or confusion, we elaborated more on the schematic details of the revised manuscript.

Finally, we thank Reviewer 1 again for many constructive and extremely important suggestions and criticisms, which we believe greatly improved our manuscript.

DETAILED REPLY TO REVIEWER 2

We thank Reviewer 2 for the important comments and suggestions which significantly improved our manuscript. Blue parts are from the referee comments.

The authors report ultra-fast rectification of the THz radiation on thin ring-shaped triangular tunnel junctions created at the interface between two gold regions. Coupling of the gold film endowed with such junctions to impinging picosecond pulse of the electromagnetic radiation induces Eddie currents in the film that causes potential redistribution at the tunnel interface between the film and the gold island. In particular, the potential difference across the junction can be so strong that it ensures finite tunnelling current between the gold film and the island. Furthermore, thanks to the triangular asymmetric shape of the island the built-up potential depends on the relative orientation between the surface current direction and the vector orthogonal to the triangle sides that defines the predominant direction of the tunnel current and leading to a finite dc signal measured between the film and the island (as oppose to the symmetric square-shaped loop-junction where the total tunnel current will be cancelled by the symmetry). Moreover, the authors demonstrated the control of the tunnelling current by femtosecond optical pulse and dc bias. The approach proposed by authors to control THz fields is novel, well explained and will be of interest for Nature Communication readers. I would recommend the publication of this manuscript in Nature Communications upon addressing the following comments and questions.

We thank the reviewer for the review of our manuscript, noticing the value of our study, and for many constructive suggestions and critical comments.

1. First of all, the revision of the first figure is needed. In particular, it takes a while to realize where the tunnel current flows. It became clear only after reading the supplementary information and related papers on the fabrication of such junctions. Authors may want to revise Fig. 1b providing the illustration of the junction crosssection and demonstrate the measurements schematic such as it is done in Fig. S3.

We thank the reviewer for a critical comment to improve the quality of our manuscript. In the revised manuscript, we especially made efforts to upgrade the schematic details of Fig. 1 to make it more comprehensible.

2. In all figures, authors illustrate the triangles colour-coded from yellow to green but never address what these colours mean? Do they correspond to the potential build-up upon coupling to the picosecond pulse? If yes, then I would assume it to be different for different polarization angles θ_{pol} although it is the same in all the panels of Fig. 1c. Please elaborate.

Actually, the color of ring barriers illustrated in the previous manuscript is meaningless.

Thanks to the comment of Reviewer 2, now we noticed that it can confuse the readers. In the revised manuscript, all schematics representing the barriers are now drawn as thick black lines.

3. In Fig. 2b authors report the tunnel current induced by gating with femtosecond optical pulse when the polarization of the THz pulse is adjusted to give zero net flow. Authors provide the measurements at two symmetric positions of the optical laser spot. Is it possible to map the full loop (or even the full metallic island) in this way? Do the colours of the schematic triangle represent this distribution? If yes, please report the colour-map scalebar. What happens at other angles of the THz pulse polarization? It can be interesting for the readers to see few maps of the potential (or current) distribution if possible.

We thank Reviewer 2 for this wonderful suggestion. Indeed, we cannot agree more with the reviewer that it would be extremely interesting for the readers to see a few maps of the potential (or current) distribution also using more light polarizations if possible. Moreover, these tasks are definitely doable even within our experimental setup. It is possible to construct a tunneling current map for a ring barrier as a function of time and space. By changing the focal position of the optical pulse in the x - y plane while fixing the time delay between the THz and optical pulses, we can directly map the spatiotemporal dynamics of the THz tunneling current.

To be absolutely honest with the referee, however, we cannot obtain the full spatial mapping right now. There was a problem in our Ti:sapphire amplifier system which is used to generate THz pulses from a LiNbO₃ crystal. On April 12th in 2018, Mai-Tai (model name of our amplifier seed laser) was sent to its manufacturer Spectra-Physics (Newport corporation brand) in the U.S. for a repair and it would take a couple of months from now for us to be ready for using the laser system again and to rebuild the whole measurement setup.

Although we cannot provide the full map, we still can provide an important additional case relevant to the comment of Reviewer 2. We have measured data for cases described in the the following figure ($\varphi_{\text{pol}} = 60^\circ$ for the THz pulse and $\varphi_{\text{pol}} = 90^\circ$ for the optical pulse), which we have now included in Supplementary Fig. 5.

Panels from Supplementary Fig. 5. Experimental demonstration of the time-resolved ultrafast optical gating of the THz tunneling current as a function of the position and time delay of the optical pulse (triangle loop, side

length of 70 μm , gap size of 2 nm). Here, the incident polarization of the THz pulse is set to $\varphi_{\text{pol}} = 60^\circ$ and that of the optical pulse to $\varphi_{\text{pol}} = 90^\circ$.

Although we believe that the presented experimental data (including the new ones just described) are enough to demonstrate the new possibilities in ring-shaped quantum tunneling barriers (as also mentioned by Reviewer 2 “... if possible”), we fully agree with the reviewer that such additional pieces of information would be extremely interesting to the readers. Actually, we ourselves are now highly motivated by Reviewer 2 and are eager to do these experiments (not just on the contour but on the entire x - y plane) soonest possible in our future study when the fixed laser is delivered to us and our experimental setup is rebuilt on that.

4. In all the figures authors report the current in arbitrary units while the measurements are performed by a standard lock-in technique. Would that be possible to report the actual current flowing across the junction as it is done in fig. S3?

Since the standard electronics such as current preamplifier and lock-in amplifier cannot follow the speed of the THz pulse time trace, it is impossible to detect the actual time-dependent current with only the electronics and THz pulses. However, thanks to **(1) the tunneling nonlinearity** and **(2) the optical technique employed in our work**, it is possible to measure the absolute tunneling current when the THz pulse is at its peak intensity. We moved the relevant section of Supplementary information to the main manuscript (Fig. 4) since this information is important and deserves to be in the main manuscript. In brief, to get the peak tunneling current, we need to know the duration of the THz tunneling current pulse if we assume that the total rectified charges by the THz pulse (denoted as q_{THz} in the revised manuscript) are successfully acquired by the current preamplifier during the measurement. In this case, **(1)** most of the tunneling current is triggered near the extremum of the THz voltage pulse (illustrated also in Supplementary Fig. 3a). **(2)** Once the total rectified charge q_{THz} is quantified, the optical technique can reveal the THz tunneling current pulse width from analysis of the current pulse at different powers (see the inset of Fig. 4c, whose detailed explanation is in the main text and in Methods). Finally, the tunneling current presented in Fig. 4e is acquired by dividing q_{THz} by the half width of the current pulse.

With that respect, data in Fig. 3 have different noise level for different DC bias voltages. Authors may want to elaborate on this and explain if the data is normalized to some value or shown as-measured. At least, please provide the axis-ticks on the y-axis in all the figures.

We thank the reviewer for finding out this confusing point. The presented data are all drawn in the same scale without different normalizations. In the revised manuscript, we included the y-axis scale bars in all measured data.

The different noise levels shown in Fig. 5b are due to the unstable current flow at strong DC bias conditions. We observed that the current signals become noisy when the field strength applied by the DC bias under THz illumination reaches $\sim 0.5 - 1$ V/nm (2 - 4 V potential difference for 4 nm gap used in obtaining the results shown in Fig. 5b). This ‘threshold DC

field' strength also depends on the quality and thickness of Al₂O₃ film (thicker films are more vulnerable due to the long travel length for the electrons across the barrier). Near this threshold field, the DC current starts to fluctuate and affect the THz current measurement, which makes the noise shown in Fig. 5b. We added this discussion to the main text.

5. In the final part of the paper, the authors report the modulation of the tunnelling current upon changing the DC bias voltage across the junction. In Fig. 3 a and b the authors provide the green dashed line which “denotes” the THz voltage profile across the barrier? How is this curve obtained?

We thank the reviewer to point out the ambiguity of our previous explanation. The relation between THz ‘voltage’ (or THz ‘potential difference’) vs. time across the gap is acquired from the original THz pulse profile and Eq. 1. The voltage applied across the gap is a result of the charge across the gap accumulated by the effective surface current \mathbf{K} induced from the incident field. By integrating the current pulse profile ($\mathbf{K} \sim \hat{\mathbf{z}} \times \mathbf{H}_0$, where \mathbf{H}_0 is the magnetic field over the metal surface) over time (Eq. 1), the applied voltage curve across the gap can be described as,

$$V(t) \propto \int_{-\infty}^t H_{\text{inc}}(t') dt'$$

where H_{inc} is the incident magnetic field strength which is proportional to the $H_0 \approx 2H_{\text{inc}}$. We added a detailed explanation on the THz voltage pulse profile to the revised manuscript and this figure to Supplementary Information as Supplementary Fig. 1.

Panels from Supplementary Fig. 1. Left column shows original THz pulse, and right column shows the transformed THz gap voltage profile by integrating left curves in time.

6. Please provide the description of insets in Fig. 3 (green wave-like schematic).

We thank the reviewer for pointing out the required but omitted explanation. In the revised manuscript legend, we separated the inset as independent panels, Figs. 5a and 5c of the revised manuscript, and added a description of these panels to the caption: the half-wave rectification and full-wave rectification.

DETAILED REPLY TO REVIEWER 3

We thank Reviewer 3 for the critical comments and questions which significantly improved our manuscript. Blue parts are from the referee comments.

This is a nice article that was quite enjoyable to read. The subject matter is of interest to many as the rectification of high-frequency radiation is becoming a hot topic.

We thank the reviewer for the review of our manuscript, noticing the value of our study, and for many constructive suggestions and critical comments.

There are a few points that I missed in the description and measurement that I think the authors should include for a more complete paper. First, the thickness of the metal film is not listed, at least I did not notice it. This is an important parameter and should be included. If thin, the metal may not be optically opaque at either the THz or 800nm optical frequency of the two pulses.

We thank the reviewer for pointing out the lack of this important information in the main manuscript. While we had included the detailed sample dimensions in Supplementary information (100 nm thick Au film), we noticed that they are not easy to find. In the revised manuscript, we moved the sample thickness information and sample fabrication methods from Supplementary information to Methods.

Suppose that an electromagnetic wave impinges on a perfect electric conductor film (placed on the x - y plane); the incident magnetic field \mathbf{H}_{inc} induces a surface current (per unit length) $\mathbf{K} = \hat{\mathbf{z}} \times (2\mathbf{H}_{\text{inc}})$ on the film, which reflects back the incident light and blocks field smearing into the perfect conductor. For a realistic metallic film, the amount of induced current is similar to the case of a perfect conductor but now the surface current flows inside the film. The detailed response is determined from the conductivity and thickness of the metal. **(1) The metal thickness of our case, 100 nm Au film, is comparable to the skin depth (~ 170 nm) for THz waves.** In this case, the magnetic field is smeared deep inside the Au film and generates current density $\mathbf{J}(z)$ through the entire metal which decays with the distance z from the top (i.e., incidence) surface (see Fig. A below). The induced \mathbf{J} charges the gap perforated in the metal film and applies a potential difference (see Fig. B below). We can simplify the expression of \mathbf{J} by introducing an ‘effective’ surface current \mathbf{K} as

$$\mathbf{K} = \int_0^h \mathbf{J}(z) dz$$

Similar to the perfect conductor case, \mathbf{K} can be expressed by

$$\mathbf{K} \sim \hat{\mathbf{z}} \times \mathbf{H}_0$$

where \mathbf{H}_0 denotes the magnetic field outside the top metal surface. For a good metal (Au in THz frequency), we can write $\mathbf{H}_0 = 2\mathbf{H}_{\text{inc}}$ (the exact value is determined from the Fresnel coefficient) and after a detailed calculation we know that this expression is correct within a 1% error. **(2) For the 800 nm optical pulse, the metal is thicker than the skin depth (~ 25 nm).**

In this case, we can also use the effective surface current per length, \mathbf{K} , to explain the time-dependent potential across the gap. The induced current density \mathbf{J} now flows mostly through the topmost part of the Au film. When the induced current reach the gap, it dumps charges along the gap edge (see Fig. C below). Similar to case (1), the surface current can be described by $\mathbf{K} \sim \hat{\mathbf{z}} \times \mathbf{H}_0$ while the error is also within 1% if we put $\mathbf{H}_0 = 2\mathbf{H}_{\text{inc}}$.

Figure 5 of reference 26, Kang, J.-H., Kim, D.-S. and Seo, M., “Terahertz wave interaction with metallic nanostructures”, *Nanophotonics* (2018). Field enhancement in thin and thick conducting films. h : film thickness, δ : skin depth. **(A)** Magnetic field of light and current density near the thin film is drawn in red and black, respectively. The THz wave impinges on the upper side of the slab. The magnetic field is approximately twice at the incidence surface (upper side). **(B)** Case (1): The thickness is around the skin depth and there is a gap perforated in the film. The magnetic field is smeared deep inside the Au film and generates a current with density J through the metal film, which charges the gap. **(C)** Case (2): The metal film is thicker than the skin depth. The surface current with density K (J integrated inside the metal along the direction normal to the surface) now flows mostly through the upper surface of the film. When the current meets the gap, it flows down following the gap edge and charges are accumulated at the gap as the current flow decreases along the gap edge.

Second, although a reference is listed (ref. 16) where a process is shown, it is unclear in this case if the same materials are used. Is the substrate again silicon, and is the substrate left on for the measurements? If so, this should also be mentioned in the theory and numerical section as the incident half plane and back half plane (if the metal is thin) are different materials.

We thank the reviewer again for raising this important issue we missed. In our study, both silicon and quartz substrates were used. In the revised manuscript, we included the information on the substrate materials used in this work. Specifically, the data presented in Fig. 2 were obtained using a silicon substrate. All the other data presented in this work were obtained by using quartz substrates. In fact, the THz tunneling characteristic is not sensitive to the kind of substrates used, and our early experiments using only THz pulses were mostly done by using silicon substrates since they are cheaper and they serve well as a transparent medium in the THz frequency range. However, we decided to change the substrate from silicon to quartz after we conceived the optical modulation experiments since silicon absorbs a significant amount of optical pulses.

We did not have to consider the kind of substrate for calculations. The kind of materials for the incident half plane and back half plane together with the metal thickness would affect the applied field **strength** across the gap (**'amplitude'** of the time-dependent surface current, discussed in our answer to the previous question) for an incident electromagnetic field. We measured this field strength directly instead of calculating it indirectly from theory to make our analysis much more reliable. In this way, the finally applied voltage across the gap can be acquired and used in further analysis. In brief, the field enhancement factor (estimated from the transmission measurement and the incident field strength) determines the gap voltage amplitude. With this voltage value together with the sample dimension information, we can calculate the tunneling current using Simmons formula. In the revised manuscript, we moved the relevant part of Supplementary information (Supplementary Discussion 'tunneling current measurement') to Methods.

The thickness and substrate are also of interest when considering the measurement. I would like to ensure that all other factors are not factors when the rectification measurement is performed. One of these is clearly localized heating when these pulses are incident. The peak tunneling current of 0.2A (supplemental page 4) is large over such a small area as these triangle geometries and nm-scale tunnel widths, even if brief in duration. Is this temperature change effect considered?

We didn't observe any change of signals or occurrence of junction breakdown during the repeated measurements. We think that the thermal effects do not severely affect the measurements since we confirmed a quantitative agreement between the experiment and calculation neglecting thermal effects (see Fig. 4e or Supplementary Fig. 4c and 4d). The peak current density driven by the THz pulse in our experiments reached $\sim 4.3 \text{ MA/cm}^2$ at a field strength of $\sim 3.3 \text{ V/nm}$ and we found other studies that reported similar or even higher values of current densities and field strengths across the nanogap junctions without any sizable thermal effects or damages:

- 1) T. Rybka *et al.*, Sub-cycle optical phase control of nanotunnelling in the single-electron regime. *Nat Photon* **10**, 667-670 (2016).
- 2) K. Yoshioka *et al.*, Real-space coherent manipulation of electrons in a single tunnel junction by single-cycle terahertz electric fields. *Nat Photon* **10**, 762-765 (2016).
- 3) V. Jelic *et al.*, Ultrafast terahertz control of extreme tunnel currents through single atoms

on a silicon surface. *Nat Phys* **13**, 591–598 (2017).
We have added this discussion to the main manuscript.

Also, if the metal thickness is not several skin depths thick at the optical pulse wavelength of 800nm then is the effect of its absorption in the silicon considered in the device. Where would this energy go in the device under the test conditions.

As we discussed before, the use of silicon together with the optical pulse generated unwanted optical absorption at the substrate, thus affecting the tunneling current through the gap. Therefore, we used optically transparent quartz substrate for all optical-related measurements.

These are largely engineering issues rather than physics, however the paper would be stronger if addressed.

We really appreciate all the constructive comments and questions of the reviewer, which resulted in a significant improvement in the quality of our manuscript.

REVIEWERS' COMMENTS:

Reviewer #1 (Remarks to the Author):

Kang et al made substantial improvement in the revised manuscript that makes the physics and novelty of current work much clearer. They provided detail responses to my questions/comments and I'm happy with most of them. As I commented before this is a solid work that contains enough new physics and carefully done analysis. In principle I can recommend the work to Nature Communications given the two minor points can be clarified further.

1. When considering the scenario for the concurrent THz and optical field illumination, authors should also comment on how the heating effect may affect the results and analysis from the high photon energy, optical pulses. Unlike THz pulses that E field acceleration is the main effect, optical field may induce strong local heating especially consider the tight focus. Such effect could smear out the nonlinearity. It will be beneficial for authors to further comment on this.

2. Authors added "... the non-vanishing total

current through the barrier naturally emerges for the triangle shape because of the nonlinearity in tunneling current vs. applied potential relation together with the triangle's lack of inversion symmetry."

Could authors specify the nonlinearity in the tunneling current here? Do they refer to the nonlinear current with high harmonics in frequency or some other nonlinear effects? I think such discussion will be very useful for the community to better digest the message.

Reviewer #2 (Remarks to the Author):

The authors addressed all my enquires, answered my questions and, as far as I have noticed, included additional information to the main text as requested by other referees. I observed a significant improvement of the manuscript which now fits the standards of the Nature Comm.

Reviewer #3 (Remarks to the Author):

My questions in the first review have been answered to my satisfaction. I thank the authors for clarifying these points.

I have also read the other reviewers' comments and the authors' responses. I concur with the other reviewer comments and the author revisions.

DETAILED REPLY TO REVIEWER 1

We thank Reviewer 1 for the important comments and suggestions which significantly improved our manuscript. Blue parts are from the referee comments.

=====

1. When considering the scenario for the concurrent THz and optical field illumination, authors should also comment on how the heating effect may affect the results and analysis from the high photon energy, optical pulses. Unlike THz pulses that E field acceleration is the main effect, optical field may induce strong local heating especially consider the tight focus. Such effect could smear out the nonlinearity. It will be beneficial for authors to further comment on this.

First, we did not observe any change of signals or breakdown of junctions during the repeated measurements. Moreover, the good agreement between the experimental results and the results of our calculation obtained by neglecting the heating effects (see Fig. 5c or Supplementary Fig. 4c and 4d) indicates that the local heating due to optical pulses does not affect the tunneling current much. In fact, the peak-field strengths of both the THz pulses (~ 3 V/nm) and optical pulses (~ 8 V/nm) are much lower than those used in, e.g., the following studies where it is reported that no sizable thermal effects or damages occurred.

- 1) T. Rybka et al., Sub-cycle optical phase control of nanotunnelling in the single-electron regime. *Nat. Photon.* **10**, 667-670 (2016).
- 2) K. Yoshioka et al., Real-space coherent manipulation of electrons in a single tunnel junction by single-cycle terahertz electric fields. *Nat. Photon.* **10**, 762-765 (2016).
- 3) V. Jelic et al., Ultrafast terahertz control of extreme tunnel currents through single atoms on a silicon surface. *Nat. Phys.* **13**, 591–598 (2017).

We added this discussion to the revised manuscript.

2. Authors added "... the non-vanishing total current through the barrier naturally emerges for the triangle shape because of the nonlinearity in tunneling current vs. applied potential relation together with the triangle's lack of inversion symmetry.". Could authors specific the nonlinearity in the tunneling current here? Do they refer to the nonlinear current with high harmonics in frequency or some other nonlinear effects? I think such discussion will be very useful for the community to better digest the message.

The nonlinearity in the tunneling current indicates simply the nonlinear dependence of the tunneling current on the applied voltage, i.e., nonlinear I - V characteristics. As the reviewer mentioned, this nonlinearity would necessarily result in higher-frequency harmonic tunneling currents if continuous waves are used. In our case, an optical pulse contains many optical cycles (thus the spectral weight is sharply peaked around 800 nm) and hence there must be such higher harmonic components, which however is not within the scope of this study. On the other hand, we described the THz pulses and their current responses in time domain instead of frequency domain since our THz pulses contain only one cycle or so. We revised our manuscript to clarify this point.